# Multimodal Evaluation of Long-Term Salivary Gland Alterations in Sarcoidosis

**DOI:** 10.3390/jcm11092292

**Published:** 2022-04-20

**Authors:** Benedikt Hofauer, Miriam Wiesner, Konrad Stock, Friedhelm Peltz, Felix Johnson, Zhaojun Zhu, Adam Chaker, Andreas Knopf

**Affiliations:** 1Otorhinolaryngology/Head and Neck Surgery, Klinikum Rechts der Isar, Technical University of Munich, 81675 Munich, Germany; miriam.a.wiesner@gmail.com (M.W.); felix.johnson@mri.tum.de (F.J.); yaya.zhu@gmx.de (Z.Z.); adam.chaker@tum.de (A.C.); 2Pneumology, Klinikum Rechts der Isar, Technical University of Munich, 81675 Munich, Germany; konrad.stock@tum.de; 3Nephrology, Klinikum Rechts der Isar, Technical University of Munich, 81675 Munich, Germany; friedhelm.peltz@mri.tum.de; 4Otorhinolaryngology/Head and Neck Surgery, University Medical Center Freiburg, University of Freiburg, 79085 Freiburg, Germany; andreas.knopf@uniklinik-freiburg.de

**Keywords:** sarcoidosis, head and neck, salivary glands, parotid gland, submandibular gland, sonography, elastography

## Abstract

Background: Sarcoidosis is a systemic inflammatory disease characterized by non-caseating granulomas. In addition to the lungs as classical site of affection, extrapulmonary manifestations are common, for example in the cervical lymph nodes or the salivary glands. The aim of this investigation is the analysis of the long-term course of glandular symptoms and the sonographic evaluation of long-term salivary gland changes. Material and methods: All patients with a diagnosis of sarcoidosis over a period of 20 years in the departments of otorhinolaryngology, rheumatology, and pneumology were identified. In addition to clinical examinations and functional evaluation of the salivary glands, a sonographic examination of the salivary glands was carried out. The changes in the area of the salivary glands were assessed using B-mode sonography and different elastographic methods with appropriate scoring systems. Results: A total of 76 patients were included in the study (age 35.1 ± 21.6 years). Overall, 17 patients presented with salivary gland manifestation at the time of the initial diagnosis. Of these patients, 15 received steroid therapy, 6 were also treated with another drug, and 2 patients were not treated with drugs. The time span between initial diagnosis and follow-up was 88.2 months (±83.0). At the time of the initial diagnosis, 17/17 complained of swelling of the salivary glands, 9/17 of xerostomia, and 8/17 of pain in the area of the salivary glands. At the time of follow-up, 5/17 reported swelling of the salivary glands, 6/17 reported xerostomia, and 1/17 reported pain in the salivary gland area. Sonography showed sonomorphological abnormalities of the salivary glands only in individual cases, with only mild alterations on average. Conclusion: In summary, it can be observed that patients with initial symptoms in the area of the salivary glands, such as swelling or pain, also suffer more frequently from dry mouth and eyes. In all patients, however, these symptoms regressed over time. A previous diagnosis of sarcoidosis with involvement of the salivary glands only leads to permanent abnormalities in the area of the salivary glands in individual cases.

## 1. Introduction

Sarcoidosis is an inflammatory multisystem granulomatous disease of unclear origin that affects individuals worldwide, and is pathologically characterized by non-caseating granulomas [1,2,3]. It predominantly affects younger adults, and typically manifests itself as bilateral hilar adenopathy and/or pulmonary reticular opacities—however, up to 30 % of patients present with extrapulmonary sarcoidosis [4,5]. Sarcoidosis can ultimately affect all organ systems with different degrees of severity; the most common sites of extrapulmonary disease are the skin, joints, eyes, reticuloendothelial system, musculoskeletal system, exocrine glands, heart, kidneys, and central nervous system [1]. Solitary extrapulmonary manifestations without additional lung pathologies are observed in approximately 8% of patients [6,7,8]. 

Diagnosis of sarcoidosis is usually based on a biopsy of a potentially affected organ and pathological evidence of non-caseating granulomas and the exclusion of other multisystemic granulomatous disorders. Only in few situations is the diagnosis based on the clinical presentation, e.g., Löfgren syndrome (erythema nodosum, hilar adenopathy, migratory polyarthralgia, and fever) or Heerfordt’s syndrome (uveoparotid fever: fever, parotid gland enlargement, anterior uveitis, facial nerve impairment) [6,9]. 

Sarcoidosis can also affect the salivary and lacrimal glands [10]. The typical signs for the involvement of these glands are xerostomia, keratoconjunctivitis sicca, or gland enlargement, which is similar to the clinical manifestations of Sjögren´s syndrome or IgG4-related systemic disease [11]. While on the one hand, the clinical presentation, especially of sarcoidosis and Sjögren´s syndrome, might be similar (or can also occur concomitantly), there are on the other hand clear differences—among other things, Sjögren´s syndrome is a chronic disease with a typical sonographic presentation within the salivary gland in 50–60% of patients, whereas sarcoidosis is a potentially curable or even self-limiting disease without typical sonographic changes within affected salivary glands [12,13,14]. So far, there is only little information on long-term sonographic and functional changes of the salivary glands in patients with sarcoidosis over the course of the disease. 

Aim of this study was to evaluate the salivary glands with different sonographic modalities and functional tests during the follow-up of patients with prior sarcoidosis to obtain more information on persistent damage and the recovery function of the salivary glands in this disease.

## 2. Material and Methods

### 2.1. Study Population

Patients with a diagnosis of sarcoidosis since 2000 and treated either at the department of otorhinolaryngology/head and neck surgery, the department of pneumology, or department of the nephrology at the Klinikum rechts der Isar, Technical University of Munich, Germany, were identified, and were offered the opportunity to take part in this study. Only patients for whom the diagnosis was confirmed on the basis of a histological examination of affected tissue or on the basis of a clear clinical combination of symptoms (e.g., Heerfordt´s syndrome or asymptomatic bilateral hilar adenopathy) were included in the study. Patients who were under 18 years at the time of the follow-up and who declined to take part in the follow-up evaluation were excluded. All examinations were performed from October 2017 to February 2018. The study protocol was in accordance with the Declaration of Helsinki. The Institutional Review Board of the Medical Faculty, Technical University of Munich, reviewed and approved the protocol (468/18). Written informed consent was obtained from all participants prior to the begin of the examination.

### 2.2. Clinical Parameter

Relevant symptoms (e.g., xerostomia, keratoconjunctivitis sicca, parotid gland enlargement, facial nerve affection, uveitis) were evaluated with visual analogue scales at the baseline and at the time of follow-up. Levels of initial ACE (angiotensin converting enzyme) and sIL2-R (soluble interleukin-2 receptor) were collected. Unstimulated salivary flow (UWSF) tests and Schirmer tests were conducted to evaluate salivary and lacrimal gland function. Chest X-rays were scored according to the Scadding stages [9,15]. 

### 2.3. Sonographic Evaluation

All sonographic examinations were performed simultaneously with the other diagnostic evaluations (Acuson S2000, 9L4, Siemens Healthcare, Erlangen, Germany). B-mode sonography, real-time tissue elastography (RTE), virtual touch imaging (VTI), and virtual touch quantification (VTQ) were performed on both parotid and submandibular glands. In VTQ, every examination consisted of 10 single measurements taken at a depth of 1.0 cm in the centre of the caudal pole of the parotid gland and the centre of the submandibular gland (to guarantee a maximum distance to the jawbone or adjacent blood vessels) within a preferably homogeneous area of the gland. The mean value of the 10 single measurements was calculated and used for further analysis. Measurements were obtained with moderate transducer pressure timed to the absence of swallowing manoeuvres. To avoid circadian variation, all examinations were performed at the same time during the morning. Patients were asked not to eat, drink, or smoke two hours prior to the examination. The echostructure of the parotid and submandibular glands in B-mode sonography was graded on a scale of 0 to 4, according to a previously published scoring system [16]: grade 0: normal, homogeneous gland; grade 1: mild parenchymal inhomogeneity (PIH), hypoechoic areas <2 mm; grade 2: evident PIH, hypoechoic areas of 2–6 mm; grade 3: gross PIH, hypoechoic areas >6 mm; grade 4: adipose degeneration of the gland, adipose tissue echogenicity, and parenchymal atrophy. The B-mode result was scored as abnormal if the score was 2 or higher, which had proved to be the optimal cut off in previous studies. RTE generated colour-coded elastogram maps (ranging from purple = soft over blue and green to red = hard), and was graded on a scale of 1 to 3. This scoring system had been adapted from a similar scoring system in patients with liver fibrosis: pattern 1: diffuse soft pattern, homogeneously spread, light-green coloured; pattern 2: intermediate pattern, partially mottled, dotted image with blue spots on a light green background; pattern 3: patchy hard pattern, mixed images with a patchwork effect of light green, red, and blue. The RTE results were rated as abnormal if the score was 2 or higher. The quality factor in RTE had to be larger than 75%. VTI generated grey scale-coded elastogram maps (ranging from white = soft over increasing grey levels to black = hard), and was also graded on a scale of 1 to 3: pattern 1: homogeneously spread, a white or light grey pattern; pattern 2: intermediate, partially mottled, a grey pattern; pattern 3: patchy, mixed images with a mostly darker grey and black pattern. VTI results were rated as abnormal if the score was 2 or higher. In VTQ, absolute values are generated; therefore no additional scoring is required [12].

Sonographic images representing the average echogenicity of the examined salivary gland were archived.

### 2.4. Statistical Analysis

Statistical analysis was performed using version 28.0 of the Statistical Package for Social Sciences software (SPSS, Chicago, IL, USA). Descriptive data are reported as mean ± standard deviation, if not otherwise stated. The normal distribution of variables was tested by using the Kolmogorov–Smirnov test. Paired *t*-tests were used for normally distributed variables, and the Wilcoxon test for not normally distributed variables. Fisher’s exact test was used to determine whether or not there is a significant association between two categorical variables. *p*-values of less than 0.05 were considered as statistically significant. 

## 3. Results

### 3.1. Study Population

Of the patients who were identified with a diagnosis of sarcoidosis, 76 could be included in the further investigations. Baseline details on the study population are illustrated in Table 1.

### 3.2. Baseline Clinical Parameter Depending on Salivary Gland Involvement

For further evaluation, the entire cohort was divided into two groups—patients with and without symptoms in the area of the salivary glands. Patients were classified as having salivary gland involvement in the case of swelling or pain of the parotid or submandibular gland at the time of diagnosis. From the entire cohort of 76 patients, 17 patients (22.4%) presented with salivary gland symptoms at baseline, from which all reported salivary gland enlargement, which was painful in 7 patients. Patients with initial salivary gland affection complained more of xerostomia and keratoconjunctivitis sicca (*p* = 0.009 and *p* = 0.014)—a significant correlation between xerostomia and salivary gland enlargement (R = 0.349, *p* = 0.002) and pain (R = 0.410, *p* < 0.001) could be shown. A comparison of other baseline characteristics, such as gender distribution, age, or serological parameters, did not reveal any differences. Further information on the differences between these two subgroups are illustrated in Table 2.

### 3.3. Multimodal Evaluation at Follow-Up

The follow-up took place 88.2 ± 83.0 months after initial diagnosis. A total of 15 out of 17 patients with salivary gland involvement received a medical treatment with steroids (43 out of 59 patients without salivary gland involvement, *p* = 0.407), and 6 out of 17 patients with salivary gland involvement received another medical treatment (17 out of 59 without salivary gland involvement, *p* = 0.773). At the time of the follow-up, 5 patients were still complaining of salivary gland enlargement, which was painful in one patient (both *p* < 0.001 compared to baseline). Other parameters are illustrated in Table 3. Patients with initial salivary gland affection did not have any persistent impairment which was different from the patients without initial salivary gland affection.

In the sonographic assessment of the salivary glands, there was also no clinically relevant difference in the assessment of the glandular parenchyma between patients with and without initial salivary gland affection for the modalities used at the time of follow-up (Table 4)—only in the acoustic radiation force impulse-based methods was a tendency towards higher values in the group with former salivary gland affection observed. Accordingly, it can be assumed that sarcoidosis does not lead to any permanent intraglandular damage or characteristic sonographic changes. Selected cases, including different appearances of the major salivary glands during the sonographic examination, are presented in Figure 1, Figure 2 and Figure 3.

During the sonographic evaluation, the number of cervical and intraparotideal lymph nodes with a size above one centimeter was evaluated. In patients with salivary gland involvement, an average of 3.2 lymph nodes (±2.1) were found, and in patients without salivary gland involvement, an average of 3.9 (±4.6, *p* = 0.561) were found.

## 4. Discussion

The current study evaluated the salivary glands in patients with prior sarcoidosis with different sonographic modalities and functional tests to obtain more information on persistent damage and sonographic alterations, as well as the potential recovery function of the salivary glands in this disease.

Overall, salivary gland involvement is a rare manifestation of sarcoidosis, and is seen in 0.5 to 5 % of cases [11,17]. Sarcoidosis in the area of the salivary glands must be suspected in uni- or bilateral enlargement predominantly of the parotid glands, but also of the submandibular glands. The swellings are usually painless. In addition, intraglandular lymph node swellings can cause the suspicion of a tumor of the parotid gland [18]. Clinically, patients can also complain of dry mouth or dry eyes. A rare manifestation is the so-called Heerfordt’s syndrome [13]. At the time of the follow-up, the symptoms in the salivary gland were clearly regressive overall. This confirms the beneficial course of sarcoidosis. Our analysis could even show that, of the 13 patients with involvement of the facial nerve at baseline, only one patient reported this impairment at the time of the follow-up. So far, there have been only a few studies of sonographic changes in the salivary glands in sarcoidosis. In a few case reports, the sonographic appearance of the affected glands has been described as hypoechogenic and hypervascular areas within an entirely heterogeneous parenchyma [19,20,21]. However, in various other reports, no specific findings are seen [19,22,23]. Previous to the presented results, there has been no information on potential remaining salivary gland damage/alteration in sonography in patients with prior sarcoidosis. We did not observe consistent remaining alterations via ultrasound in patients with initial salivary gland affection compared to patients without. In some cases, even patients without significant clinical involvement of the salivary glands showed abnormalities comparable to Sjögren’s syndrome, which may indicate either a subclinical involvement of the salivary glands or a co-existing Sjögren’s syndrome [24]. On the other hand, even in patients with a significant enlargement of the salivary glands, a regular glandular parenchyma could be shown in the follow-up.

Sjögren’s syndrome is a disease that can be very similar in its symptoms to sarcoidosis, especially when the salivary glands are involved. For this reason, a known sarcoidosis in some classification criteria is also an exclusion criterion for the classification of Sjögren’s syndrome [25]. Sjögren’s syndrome, a systemic autoimmune disease, leads to damage of the exocrine glands, and, consequently, to dry mouth and eyes. Especially in the initial phase of the disease, the swelling of the salivary glands can be part of the clinical appearance [26]. Sjögren’s syndrome is associated with characteristic sonographic changes in the salivary glands. The majority of patients show either hypoechogenic lesions or hyperechogenic bands in the glandular parenchyma. Although there have been few studies on changes in these abnormalities over time so far, they appear to be stable, and cannot be significantly influenced by therapeutic interventions. The same applies to the accompanying dry mouth and eyes, which cannot be reversed, and are only treated symptomatically [27]. This represents a decisive difference to sarcoidosis. Here, either spontaneously or through drug therapy, a complete improvement of the symptoms and a cure is possible. It is therefore not surprising that the ultrasound examination of the salivary glands does not reveal any permanent damage.

Another possible reason as to why only few abnormalities could be documented in B-mode sonography is the evaluation system that was used. In the meantime, there are various evaluation systems used specifically for Sjögren’s syndrome, which include various changes in the parenchyma, and form a score from them [28,29]. The rating system used was also adapted from a rating system for Sjogren’s syndrome [16]. It cannot be ruled out that some changes were not fully recorded as a result. When looking through all the documented images, however, no further characteristic changes were observed.

Various limitations of this study need to be addressed. The multimodal follow-up was carried out in all patients at different times in relation to the initial diagnosis, and this impaired the comparability of the results. Furthermore, there is insufficient information on sonographic changes at the time of the initial diagnosis, therefore it cannot be said with certainty whether sonographic changes have regressed or never existed. To overcome this limitation, further prospective studies on sonographic alterations should follow—a multi-center design can help to generate larger numbers of cases in a shorter time.

In summary, it can be observed that patients with initial symptoms in the area of the salivary glands, such as swelling or pain, also suffer more frequently from dry mouth and eyes. In all patients, however, these symptoms regressed over time. A past sarcoidosis diagnosis with involvement of the salivary glands only leads to permanent abnormalities in the area of the salivary glands in individual cases. This is important information for the differential diagnosis in patients with condition after a sarcoidosis.

## Figures and Tables

**Figure 1 jcm-11-02292-f001:**
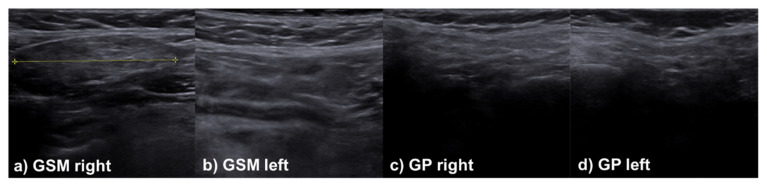
Case of a female patient, 52-year-old at the time of diagnosis, presenting with a mediastinal lymphadenopathy as an initial sign of sarcoidosis. Despite having no complaints in the area of the salivary glands at baseline (no enlargement, pain or xerostomia), all salivary glands (GSM = submandibular gland, GP = parotid gland) are inhomogeneous with hyperechogenic bands, similar to Sjögren’s syndrome.

**Figure 2 jcm-11-02292-f002:**
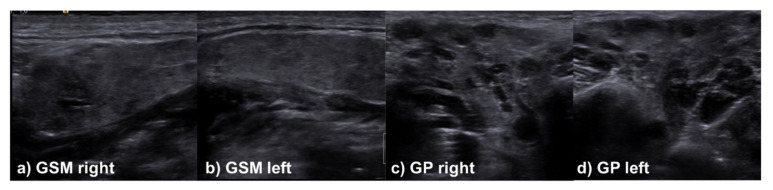
Case of a 36-year-old male patient presenting with recurrent swelling of both parotid glands and submandibular gland to a lesser extent without xerostomia. Both parotid glands show a clear inhomogeneity, and are interspersed with multiple hypoechogenic areas. Both submandibular glands have a slightly altered homogeneity with faint hypoechogenic lesions.

**Figure 3 jcm-11-02292-f003:**
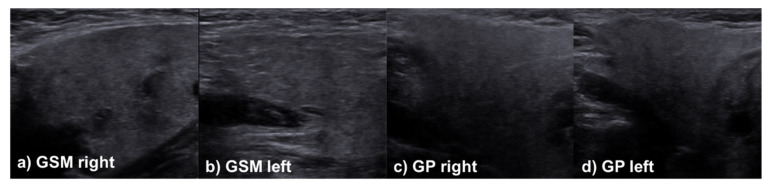
Case of a 30-year-old female patient presenting with Heerfordt’s syndrome (including enlargement of both parotid glands, anterior uveitis, fever, and palsy of the facial nerve). While all symptoms have completely resolved, the patient is still complaining about an indolent swelling of all salivary glands, even 10 years after initial diagnosis. In B-Mode sonography, normal salivary glands are seen, without any alterations within the parotid glands.

**Table 1 jcm-11-02292-t001:** Baseline details on the study population (*n* = 76).

Gender distribution [% female]	65.8
Age at diagnosis [years]	35.1 ± 21.6
Treatment department [n/%]	
*ENT*	23/30.3
*Rheumatology*	27/35.5
*ENT + Rheumatology*	15/19.7
*Pneumology*	11/14.5
ACE [U/L]	85.3 ± 59.1
sIL-2R [U/mL]	1461.3 ± 959.4
Chest X-ray [n/%]	
*Normal*	6/7.9
*Bihilar lymphadenopathy*	26/34.2
*Bihilar lympha. with lung involvement*	27/35.5
*Lung involvement without bihilar lympha.*	4/5.3
*Lung fibrosis*	0/0
ENT symptoms [n/%]	60/78.9
Arthralgia [n/%]	14/18.4
Uveitis [n/%]	14/18.4
Skin involvement [n/%]	27/35.5
Fever [n/%]	17/22.4

**Table 2 jcm-11-02292-t002:** Baseline clinical parameter depending on salivary gland involvement (entire cohort *n* = 76) (VAS = visual analogue scale).

Parameter	With Salivary Gland Involvement (*n* = 17)	Without Salivary Gland Involvement (*n* = 59)	*p*-Value
Gender distribution [% female]	58.8	67.9	0.495
Age at diagnosis [years]	36.6 ± 9.9	34.71 ± 24.0	0.762
Interval diagnosis-follow-up [months]	88.2 ± 73.4	88.3 ± 87.2	0.997
ACE [U/L] #	103.3 ± 68.3	79.0 ± 55.6	0.294
sIL-2R [U/mL] §	1059.0 *	1494.83 ± 994.1	0.682
Xerostomia [VAS 0–10]	4.5 ± 4.6	2.1 ± 2.9	0.009
Keratoconjunctivitis sicca [VAS 0–10]	4.5 ± 3.8	2.2 ± 3.1	0.014
Uveitis [n/%]	5/29.4	8/13.6	0.129
Facial nerve palsy [n/%]	5/29.4	8/13.6	0.129
Fever [n/%]	5/29.4	11/18.6	0.358

# Normal values of ACE: 8–62U/l; § normal values of sIL-2R: 112–502U/mL; * value only available for one patient.

**Table 3 jcm-11-02292-t003:** Clinical parameter depending on salivary gland involvement at follow-up (entire cohort *n* = 76).

Parameter	With Salivary Gland Involvement (*n* = 17)	Without Salivary Gland Involvement (*n* = 59)	*p*-Value
Schirmer test [mm]	20.0 ± 11.2	15.9 ± 9.8	0.144
UWSF [mL/min]	1.3 ± 0.9	1.2 ± 1.1	0.706
Xerostomia [VAS 0–10]	2.7 ± 3.8	2.4 ± 3.3	0.760
Keratoconjunctivitis sicca [VAS 0–10]	3.5 ± 3.8	2.4 ± 3.2	0.241
Uveitis [n/%]	2/11.8	3/5.1	0.310
Facial nerve palsy [n/%]	1/5.9	0/0	0.224
Fever [n/%]	0/0	1/1.7	0.776

**Table 4 jcm-11-02292-t004:** Sonographic evaluation depending on salivary gland involvement at follow-up (entire cohort *n* = 76) (RTE = real-time tissue elastography, VTI = virtual touch imaging, VTQ = virtual touch quantification).

Parameter	With Salivary Gland Involvement (*n* = 17)	Without Salivary Gland Involvement (*n* = 59)	*p*-Value
B-Mode all glands [Δ score 0–4]	0.5 ± 0.6	0.4 ± 0.5	0.523
B-Mode parotid glands [Δ score 0–4]	0.5 ± 0.8	0.3 ± 0.7	0.106
RTE all glands [Δ score 1–3]	3.0 ± 0.4	3.0 ± 0.1	0.502
RTE parotid glands [Δ score 1–3]	2.9 ± 0.2	3.0 ± 0.1	0.697
VTI all glands [Δ score 1–3]	2.1 ± 0.4	1.9 ± 0.4	0.041
VTI parotid glands [Δ score 1–3]	2.4 ± 0.5	2.2 ± 0.5	0.082
VTQ all glands [m/s]	1.9 ± 0.4	1.7 ± 0.3	0.235
VTQ parotid glands [m/s]	2.4 ± 0.5	2.2 ± 0.5	0.082

## Data Availability

The data presented in this study are available on request from the corresponding author.

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
