# Peer review of "Multimodal Evaluation of Long-Term Salivary Gland Alterations in Sarcoidosis"

_jcm, 2022, doi:10.3390/jcm11092292_

Round 1

Reviewer 1 Report

more well defined inclusion and exclusion criteria is needed

Author Response

Thank you very much for your comment, I included further information on in- and exclusion criteria in the M&M section.

Reviewer 2 Report

The authors wrote an article about multimodal evaluation of long-term salivary gland alterations in sarcoidosis. The article in very interesting, well written and the topic was not really discussed in literature. The article is worthy of publication, but there are some corrections to do to improve the scientific impact of the manuscript.

  1. In the statistical analysis, in the Table 4, Uveitis [n/%] Facial nerve palsy [n/%] Fever [n/%], you can not use the "t test" to compare it. You should use the Fisher exact test. Please add this in the statistical part.
  2. You should increase the limit session in the discussion. For example, the limit of simple size and, how future prospective, the possibility to do a multicenter study.
  3. You should not only write the age of diagnosis, but the time from the first diagnosis in the table 3.
  4. Please, talk more about the possibility of surgery for salivary gland in these cases, looking for other articles in literature. Never forget the Frey syndrome in case of Surgery. Using these reference, please talk more about the existence of Mikulicz's syndrome, that was included in Sjogren disease: Freni F, Gazia F, Stagno d'Alcontres F, Galletti B, Galletti F. Use of botulinum toxin in Frey's syndrome. Clin Case Rep. 2019 Jan 31;7(3):482-485.

Author Response

Thank you very much for your constructive comments.

  1. In the statistical analysis, in the Table 4, Uveitis [n/%] Facial nerve palsy [n/%] Fever [n/%], you can not use the "t test" to compare it. You should use the Fisher exact test. Please add this in the statistical part.

Thank you very much for this comment, I recalculated this and inserted both the new p-values as well as an information on this into the statistical part. 

  1. You should increase the limit session in the discussion. For example, the limit of simple size and, how future prospective, the possibility to do a multicenter study.

A comment on this was added into the limitation-section in the discussion.

  1. You should not only write the age of diagnosis, but the time from the first diagnosis in the table 3.

I added this information into the table (now table 2).

  1. Please, talk more about the possibility of surgery for salivary gland in these cases, looking for other articles in literature. Never forget the Frey syndrome in case of Surgery. Using these reference, please talk more about the existence of Mikulicz's syndrome, that was included in Sjogren disease: Freni F, Gazia F, Stagno d'Alcontres F, Galletti B, Galletti F. Use of botulinum toxin in Frey's syndrome. Clin Case Rep. 2019 Jan 31;7(3):482-485.

Thank you very much for this comment. The treatment of sarcoidosis is usually not based on surgical interventions but more on medical approaches, if indicated. Therefor after discussion within the team we did not any further comments on surgeries of the salivary glands or potential complications of these surgeries. However, we thank the reviewer for the interesting article and we will keep it in mind for future projects. 

Reviewer 3 Report

direct vocabulary assessment

1-ex- pulmunology - pneumology - igurepulmunology - so whats the department name??

2-why there was no control group with healthy glands/nodes to examine and correlate the data?

3-results and disussion section should be improved

4-does ACE adress some more data?

5-does patient age correlate with the results?

6-does the occurence or any other extra nodal, extra-glandular sarcoidosis is important for clinicians?

Author Response

Thank you very much for your constructive comments.

direct vocabulary assessment

Thank you, we assessed the manuscript regarding language. 

1-ex- pulmunology - pneumology - igurepulmunology - so whats the department name??

We unified the spelling to pneumology, thank you for this observation.

2-why there was no control group with healthy glands/nodes to examine and correlate the data?

Thank you for this comment. We did not include a control group with healthy volunteers because we wanted to focus on the longitudinal changes within the patient population and not the comparison between patients with sarcoidosis and healthy volunteers at a specific time point. 

3-results and disussion section should be improved

Thank you for this comment. We modified the illustration of some of the results, included some new results into the tables, recalculated some p-values and added some information into the discussion (as you can see in the track changes of the manuscript).

4-does ACE adress some more data?

ACE values were collected at the time of the initial diagnosis and in both patients with initial salivary gland symptoms and in patients without initial salivary gland symptoms we observed elevated values. Normal values for ACE (and sIL-2R) have been added.

5-does patient age correlate with the results?

No, we did these calculations and age did not correlate with any findings. 

6-does the occurence or any other extra nodal, extra-glandular sarcoidosis is important for clinicians?

Yes, absolutely. In cases of sarcoidosis a clinical examination should take place to evaluate, which organ systems are affected by the disease. This could have an impact on the decision how to treat the patient.

Round 2

Reviewer 3 Report

thank you

best luck with future studies